# Galectin-9 Triggers Neutrophil-Mediated Anticancer Immunity

**DOI:** 10.3390/biomedicines10010066

**Published:** 2021-12-29

**Authors:** Natasha Ustyanovska Avtenyuk, Ghizlane Choukrani, Emanuele Ammatuna, Toshiro Niki, Ewa Cendrowicz, Harm Jan Lourens, Gerwin Huls, Valerie R. Wiersma, Edwin Bremer

**Affiliations:** 1Department of Hematology, University Medical Center Groningen (UMCG), University of Groningen, 9713 GZ Groningen, The Netherlands; n.ustyanovska.avtenyuk@umcg.nl (N.U.A.); g.choukrani@umcg.nl (G.C.); e.ammatuna@umcg.nl (E.A.); e.krol@umcg.nl (E.C.); h.j.lourens@umcg.nl (H.J.L.); g.huls@umcg.nl (G.H.); 2Department of Immunology, Kagawa University, Takamatsu, Kagawa 760-0016, Japan; niki.toshiro@kagawa-u.ac.jp

**Keywords:** carcinoma, galectin-9, neutrophils, trogocytosis, CD47

## Abstract

In earlier studies, galectin-9 (Gal-9) was identified as a multifaceted player in both adaptive and innate immunity. Further, Gal-9 had direct cytotoxic and tumor-selective activity towards cancer cell lines of various origins. In the current study, we identified that treatment with Gal-9 triggered pronounced membrane alterations in cancer cells. Specifically, phosphatidyl serine (PS) was rapidly externalized, and the anti-phagocytic regulator, CD47, was downregulated within minutes. In line with this, treatment of mixed neutrophil/tumor cell cultures with Gal-9 triggered trogocytosis and augmented antibody-dependent cellular phagocytosis of cancer cells. Interestingly, this pro-trogocytic effect was also due to the Gal-9-mediated activation of neutrophils with upregulation of adhesion markers and mobilization of gelatinase, secretory, and specific granules. These activation events were accompanied by a decrease in cancer cell adhesion in mixed cultures of leukocytes and cancer cells. Further, prominent cytotoxicity was detected when leukocytes were mixed with pre-adhered cancer cells, which was abrogated when neutrophils were depleted. Taken together, Gal-9 treatment potently activated neutrophil-mediated anticancer immunity, resulting in the elimination of epithelial cancer cells.

## 1. Introduction

Galectin-9 (Gal-9) is a β-galactoside-binding galectin that comprises two homologous distinct carbohydrate recognition domains (CRDs) connected by an inter-domain linker [1]. Gal-9 is involved in many biological processes in the human body including the regulation of epithelial cell polarity and cell migration, but it is also involved in malignant progression [2]. For instance, the loss of Gal-9 expression is closely associated with tumor progression and metastasis formation in various epithelial cancers [3,4,5]. Further, (re)expression of Gal-9 in cells or treatment with recombinant Gal-9 suppresses cell proliferation and tumor growth in various cancers including breast cancer [3], hepatocellular carcinoma [6], cholangiocarcinoma [7], and colon cancer [8]. This cytotoxicity in colon cancer is independent of caspases and characterized by autophagosome accumulation and lysosomal swelling [9]. In addition to direct cytotoxic effects, Gal-9 binds to CD44 expressed on cancer cells and, thereby, hampers metastatic spread and reduces the number of lung metastases [10]. In line with this, expression of Gal-9 in breast cancer cells inhibits adhesion to endothelium and/or extracellular matrix components such as collagen type I [10,11]. Based on these studies, loss of Gal-9 during cancer progression was hypothesized to modulate cell adhesion and, thereby, facilitate extravasation and metastatic spread. 

Gal-9 is perhaps even better known as a multifaceted player in innate and adaptive immune responses with the first report on Gal-9 detailing its function as an eosinophil attractant [12]. Indeed, Gal-9 is critical for immune homeostasis with, for example, impaired mucosal immunity in Gal-9 knockout mice and a pivotal alarmin function during sepsis in mice [13,14]. Gal-9 is further linked to immune disbalance in patients with elevated serum levels of Gal-9 associating with disease activity in rheumatoid arthritis [15], type 2 diabetes, and chronic kidney disease [16]. Moreover, Gal-9 impacts on antitumor immunity, e.g., by activation of antigen presentation by dendritic cells that subsequently drives anticancer CD8 T-cell immune responses [17].

Gal-9 binds to and modulates activity of many if not all immune cell subtypes with binding to T cell immunoglobulin mucin domain-containing protein 3 (TIM-3) on T cells reportedly inducing cell death of T helper type 1 cells, suppressing TH17 cells, and promoting differentiation of regulatory T cells (Tregs) [18]. In contrast, we and others demonstrated that recombinant Gal-9 also potentiates signaling in T cells independent of TIM-3, with a shift from naive cells towards central memory and INFγ producing Th1 phenotype [19]. In a further study, T-cell activation was related to Gal-9-mediated calcium mobilization through TCR–CD3 complexes and the tyrosine kinase Lck [20]. Gal-9 also acts on innate immune cells by inducing maturation of monocyte-derived dendritic cells and triggering pro-inflammatory cytokine production by monocytes [21]. Further, Gal-9 stimulates the differentiation of human macrophages towards an M2 phenotype macrophage in vitro [22,23], whereas in murine macrophages, the upregulation of endogenous Gal-9 associates with an M1 phenotype [24,25]. Moreover, we recently identified Gal-9 as a prominent activator of neutrophils ex vivo that, among others, increased neutrophils’ life span, upregulated integrins, and induced secretion of the pro-inflammatory cytokine IL-8 [26]. This Gal-9-mediated activation of neutrophils in the context of rheumatoid arthritis triggered pathogenic post-translational modification of proteins [15].

Of note, during metastatic spread into the blood stream, the main cell type encountered by cancer cells are the polymorphonuclear cells (PMNs), of which ~95% are neutrophils. These innate immune cells are the first line of defense in the immune response and, although best known as pro-tumorigenic and possibly pro-metastatic, also have documented anti-tumoral activity [27,28]. Interestingly, we previously identified that treatment of cancer cells with Gal-9 triggers the upregulation of phosphatidylserine (PS) exposure [5,9], an important “eat me” signal involved in the phagocytic balance. Indeed, PS exposure is critical for homeostatic removal of dying/superfluous cells [29,30]. Thus, Gal-9-treated cancer cells may become sensitive to elimination by phagocytes.

In the current study, we identified that Gal-9 on the one hand sensitized carcinoma cells to immune recognition by increasing PS exposure within minutes as well as rapidly decreasing surface expression of the key “don’t eat me” signal CD47. Treatment with Gal-9 also prominently activated PMNs (termed neutrophils in the remainder of the manuscript), leading to cancer cell trogocytosis as well as neutrophil-mediated cytotoxic activity in cancer cells. Taken together, Gal-9 potently activated neutrophil-mediated anticancer immune responses that might play a role in hampering metastatic dissemination in the bloodstream.

## 2. Materials and Methods

### 2.1. Galectin-9 and Inhibitors

Recombinant Gal-9 (rGal-9) (also known as Gal-9(0)) containing a truncated 2 amino acid inter-domain linker was produced as described before [30]. The following inhibitors were used: α-lactose (Sigma-Aldrich, St. Louis, MO, USA), zVAD-fmk (R&D Systems, Inc., FMK001, Wiesbaden, Germany), Bapta (CALBIOCHEM, Merck KGaA, Darmstadt, Germany), EDTA (Sigma-Aldrich, Germany), and R406 (Axon MedChem).

### 2.2. Cell Lines

FaDu, DLD-1, HCT116, HT29, Caco-2, HCT-8, and WiDr were obtained from American Type Culture Collection. Cell lines were cultured either in RPMI or DMEM (Lonza, Biowhittaker BE12–604F and BE12–155F) supplemented with 10% fetal calf serum (FBS, Gibco™ Fetal Bovine Serum, USA) at 37 °C in a humidified 5% CO_2_ atmosphere.

### 2.3. Isolation of Immune Cells

Total neutrophils/white blood cell (WBC) populations were isolated by ammonium chloride lysis solution purchased from Sanquin. All donors gave informed consent (Sanquin Blood Supply, Groningen, The Netherlands). To isolate monocytes and neutrophils, peripheral blood mononuclear cells (PBMCs) were isolated by the density gradient Lymphoprep™ according to the manufacturer’s recommendations (STEMCELL Technologies, Vancouver, Canada). Cell pellets obtained after the lymphoprep procedure were lysed using ammonium chloride to obtain isolated neutrophils. To generate macrophages, PBMCs were seeded in a 6-well plate and incubated for 16 h to let the monocytes adhere. Monocytes were differentiated to macrophages (M0) in RPMI 1640 culture medium + 10% FBS supplemented with GM-CSF (50 ng/mL) and M-CSF (50 ng/mL) for 7 days. To generate M1-type macrophages, M0 cells were primed with LPS (100 ng/mL) and IFN-γ (50 ng/mL) for an additional 24 h. To generate type 2 macrophages, M0 cells were primed with IL-10 (50 ng/mL) and TGF-β (50 ng/mL) for an additional 48 h and were subsequently harvested. Assays were repeated with a minimum of three independent donors to obtain experimental replicates. 

### 2.4. Expression of PS and CD47 on FaDu

FaDu cells (5 × 10^4^/FACS tube) were pre-incubated with/without the pan-caspase inhibitor zVAD-fmk (10 μM) for 1 h, whereupon Gal-9 (50, 100, 150, and 300 nM) with/without α-lactose (40 mM, Sigma-Aldrich) was added for 1–6 h. Cells were then stained with Annexin V-FITC (ImmunoTools) in calcium buffer or anti-CD47-FITC (BioLegend, Clone CC2C6, San Diego, CA, USA) with isotype control.

### 2.5. Detection of PS Flip-Flop Using NBD-Labeled PS

The protocol was based on a previous study [31]. In brief, FaDu cells (5 × 10^4^/FACS tube) in 0.2 mL DMEM + 10% FBS were treated with Gal-9 for 1 h at 37 °C. Following the incubation period, cells were washed twice and resuspended in Hanks’ Balanced Salt Solution (HBSS) with 1 mM CaCl_2_. NBD-labeled PS (1-palmitoyl-1-(6-[(7-nitro-2–1, 3-benzoxadiazol-4-yl)aminocaproyl]-sn-glycero-3-phosphoserine) (Avanti Polar Lipids, Alabaster) was prepared by drying 5 µL in a glass tube and resuspended in 50 µL of 0.25% BSA with 200 mM phenylmethylsulfonyl fluoride in PBS. After 1 h incubation with the lipid suspension, the cells were resuspended in 0.5% BSA for an additional 5 min at RT in order to remove the probe lipids that were not introduced into the outer leaflet of the cell plasma membrane. Subsequently, the samples were then washed with 2 mL ice-cold PBS and analyzed using a CytoFLEX flow cytometer (Beckman Coulter, Indianapolis, USA).

### 2.6. Flow Cytometry-Based Trogocytosis Assay

All in vitro trogocytosis assays reported here were performed by co-incubating fluorescently-labeled target cells and neutrophils at a ratio of 1:1 overnight in a humidified, 5% CO_2_ incubator at 37 °C in FACS tubes (Greiner Bio-One, Kremsmünster, Austria) in RPMI + 10% FBS. Tumor cells were pre-stained with either 1,10-dioctadecyl-3,3,30,30–tetramethylindodicarbocyanine (DiD) or carboxyfluorescein succinimidyl ester (CFSE) following the manufacturer’s instructions. After gating for the neutrophil population based on SSC vs. FSC as well as CD11b positive macrophages, the percentage of CFSE- or DiD-positive phagocytes were evaluated. For Syk kinase inhibition, the mix population were pre-treated with 5 µM R406 for 1 h at 37 °C before Gal-9 treatment. Trogocytosis was analyzed by flow cytometry.

### 2.7. Antibody-Dependent Cellular Phagocytosis (ADCP)

FaDu cells (1 × 10^6^/mL) in 1 mL DMEM + 10% FBS were pre-incubated in a FACS tube with/without 300 nM Gal-9 for 1 h in a humidified, 5% CO_2_ incubator at 37 °C. Subsequently, cells were then washed with PBS and suspended in DMEM + 10% FBS with/without 40 mM α-lactose. Mixed cultures of neutrophils and FaDu or Gal-9-pre-treated/untreated FaDu cells were incubated at different concentrations of CTX with/without α-lactose for 16 h. The level of neutrophil-mediated phagocytosis was measured as described above.

### 2.8. Neutrophil Activation Assays

For calcium flux, isolated neutrophils (6 × 10^6^) were incubated in 0.5 mL alpha-mem (Lonza, Belgium) + 2.5% FBS supplemented with 8 nM Fluor-4, AM, cell permeant (Life Technologies, ref. 14201) at 37 °C for 1 h. Cells were washed with alpha-mem and incubated for 30 min at RT in the dark, then washed twice with PBS and resuspended in 0.5 mL HEPES buffer and treated. Neutrophils were pre-incubated with EDTA (5 µM) and/or BAPTA (50 µM) were incubated with neutrophils for 30 min at 37 °C before treatment. Calcium release was then quantified in neutrophils with/without Gal-9 treatment (300 nM) or thapsigargin (10 µM) as a function of time by flow cytometry. 

For measuring activation markers, leukocytes or isolated neutrophils (5 × 10^5^) were incubated in 1 ml RPMI + 10% FBS with/without Gal-9 (0, 15, 50, 100, and 150 nM) and in the presence of absence of α-lactose for 1 h. After incubation, Fc receptors were blocked with human FcR blocking reagent (Miltenyi Biotec) for 5 min at RT. Cells were then stained with anti-CD11b-FITC, anti-CD18-APC, anti-CD11a-FITC, anti-CD11c-FITC, anti-CD15-BV605, anti-CD66b-FITC, anti-CD63-FITC, anti-62L-FITC (ImmunoTools) at 1:100 v/v for 60 min at 4 °C. Subsequently, the stained cells were washed twice with PBS and then analyzed by flow cytometry.

### 2.9. Cell Killing Assay

For microscopy-based killing assays, tumor cells were stained with CFSE (Invitrogen™, Thermo Fisher) according to protocol and seeded at a density of 5 × 10^4^ or 10 × 10^4^/well in 96-well F-bottom plates (Corning). After washing with RPMI + 10%, freshly isolated WBCs were added at an E:T ratio of 10:1, and the mix culture was treated with different concentrations of Gal-9 (0, 15, 50, 100, and 150 nM). Then, cell killing was analyzed using the IncuCyte (S3 Live-Cell Analysis System, Ann Arbor, MI, USA) by imaging at 3 h intervals using a 10× objective. The first image time point (reported as t = 0) was generally acquired within 30–60 min after adding the WBCs to the cancer cells. 

For luciferase-based killing assays, luciferin-transduced cell lines were allowed to pre-adhere for one day at a density of 5000 cells/well in a 96-well plate (Corning^®^ 96-well Flat Clear Bottom White Polystyrene TC-treated Microplates). Neutrophils were then added at different ratios (0:1, 2:1, 5:1, and 10:1) with/without Gal-9 (50 nM) for 48 h. Then, luciferin (GoldBio) was added to the culture medium (1/100) and incubated at 37 °C for 10 min. A maximum death control was induced by treating cells with 20% v/v ethanol. A luminescence readout was performed by a luminescence reader (Synergy, BioTek). Luminescence of the maximum death control was subtracted from all values, and cell viability was calculated as a percentage of the medium control.

### 2.10. MTS Assays to Determine Cell Adhesion

For cell adhesion assays, tumor cells were mixed with WBCs in different E:T ratios in a 96-well plate. Subsequently, mixed neutrophil/cancer cell cultures were treated for 16 h with Gal-9 (0, 15, 50, and 150 nM) and as a control in the presence or absence of 40 mM α-lactose or sucrose. Non-adhered cells were removed by gently washing with pre-warmed PBS. Then, 3-(4,5-dimethylthiazol-2-yl)-5-(3-carboxymethoxyphenyl)-2-(4-sulfophenyl)-2H-tetrazolium (MTS) (CellTiter 96 AQueous One Solution Cell Proliferation; Promega, G3580) was added to the culture medium (10% (v/v)) and incubated at 37 °C. A maximum death control was induced by treating cells with 20% v/v ethanol. An MTS readout was performed by measuring absorbance at 490 nm (VersaMax microplate reader, Molecular Devices). Absorbance of the maximum death control was subtracted from all values, and cell viability was calculated as a percentage of the medium control.

### 2.11. Statistical Analysis

Statistical analysis was performed by one-way ANOVA followed by the Tukey–Kramer post-test, by paired *t*-test using Prism software, or by linear regression using SPSS (version 23). *p* < 0.05 was defined as a statistically significant difference. Where indicated * *p* < 0.05; ** *p* < 0.01; *** *p* < 0.001; **** *p* < 0.0001. All graphs show the mean +/− standard deviation, unless stated otherwise.

## 3. Results

### 3.1. Galectin-9 Shifted the Phagocytic Balance in Cancer Cells

Although some carcinoma cell lines were resistant to direct anti-carcinoma activity of Gal-9 [9], we identified previously that all cell lines responded to Gal-9 treatment with surface-exposure of phosphatidylserine (PS), an important “eat me” signal for innate immune cells. Interestingly, treatment with Gal-9 not only rapidly increased PS exposure (Figure 1A,B), it also triggered a reduction in the expression of the “don’t eat me” signal CD47 in FaDu cells (Figure 1C). Downregulation of membrane-expressed CD47 was also rapid, with a 30% decrease within 30 min and a maximal 70% decrease after 6 h of treatment (Figure 1D). Thus, Gal-9 treatment clearly altered the phagocytic balance of FaDu cells. Such modulatory effects of Gal-9 on PS and CD47 expression were similarly detected in various other carcinoma cell lines (Figure 1E,F). Of note, CD47 downregulation was dependent on the CRD carbohydrate binding activity of Gal-9, since co-treatment with α-lactose abrogated these effects (Appendix A). 

Physiologically, the level of PS on the outer leaflet is controlled on the one hand by the flipping of PS from the outer leaflet to the inner leaflet by so-called flippases. On the other hand, so-called Scramblases non-specifically translocate PS between the inner and outer membrane leaflet [32]. During apoptotic cell death, PS-exposure on the outer leaflet is predominantly attributed to caspase-mediated inactivation of flippases [33,34]. However, co-incubation with the caspase inhibitor, zVAD-fmk, did not inhibit Gal-9-induced PS-exposure (Figure 1G,H). Interestingly, using fluorescently labeled PS (NBD-PS) to monitor the internalization of PS, a strong inhibitory effect of Gal-9 on PS internalization to the inner membrane leaflet was detected compared to the medium control (Figure 1I). Thus, treatment of cancer cells with Gal-9 rapidly shifted the phagocytic balance towards a pro-phagocytic state in cell lines sensitive as well as resistant to the direct cytotoxic activity of Gal-9.

### 3.2. Galectin-9 Triggered Neutrophil but Not Macrophage-Mediated Cancer Cell Uptake

Based on the clear shift in the expression of key phagocytic regulators, Gal-9 treatment of cancer cells was performed in mixed culture experiments with macrophages and neutrophils to assess potential phagocytic uptake of cancer cells. However, FaDu cells were only minimally phagocytosed by M1 or M2c macrophages upon treatment with Gal-9 (Figure 2A). In contrast, treatment of a mixed culture of FaDu and leukocytes triggered 50–80% of trogocytosis by neutrophils (representative plots in Figure 2B). Similar trogocytic activity of neutrophils was detected for a panel of six carcinoma cell lines with cell lines resistant to direct anti-carcinoma activity of Gal-9 being equally susceptible as the sensitive cell lines to direct Gal-9 activities (Figure 2C). Trogocytosis of cancer cells by neutrophils was an active cellular process and only detected at 37 °C and not at 4 °C, as illustrated for FaDu cells (Figure 2D), was abrogated by α-lactose co-incubation (Figure 2E) and blocked upon inhibition of key trogocytosis kinase Syk (Figure 2F). Uptake of cancer cells by neutrophils upon Gal-9 treatment was rapid and detected within 30 min of treatment with further increases up to ~80% after 24 h of treatment (Figure 2G). This timeframe of trogocytosis coincided with the shift in the balance between the pro- and anti-phagocytic signals described.

### 3.3. Galectin-9 Potentiated Antibody-Dependent Cellular Trogocytosis (ADCP)

Neutrophils have previously been reported to contribute to FcγR-mediated trogocytosis upon treatment with therapeutic antibodies [35]. To assess whether Gal-9 might improve neutrophil-mediated ADCP, FaDu cells were co-treated with epidermal growth factor receptor (EGFR)-targeted therapeutic antibody cetuximab (CTX) and Gal-9, leading to strong clustering of neutrophils on top of FaDu cells and a stretched-activated phenotype of neutrophils (Figure 2H, bottom-right panel). A similar neutrophil clustering and phenotype was detected upon Gal-9 or CTX treatment alone with resting and round neutrophils only being detected in untreated conditions (Figure 2H). Notably, when quantifying trogocytosis, CTX treatment of Gal-9-pre-treated FaDu cells yielded a significant increase in trogocytosis compared to CTX treatment alone with an up to ~40% increase (Figure 2I, grey diamonds) as confirmed by linear regression analysis (β = 0.664, 95% CI 19.9–47.8, *p* < 0.001). When possible residual surface-bound Gal-9 was removed with excess α-lactose, trogocytosis of Gal9-pre-treated FaDu cells still increased by ~20% (Figure 2J). Similar potentiating activity was detected using DLD-1 colon carcinoma cells (Appendix A). Together, these data indicate that Gal-9 actively changed the immunological balance towards cancer cell removal, leading to trogocytosis that potentiated antibody-mediated activity.

### 3.4. Gal-9-Mediated Trogocytosis Can Largely Be Attributed to Neutrophil Activation

As reported above, Gal-9 triggered a prominent change in the phagocytic balance on cancer cells, with downregulation of CD47 and PS exposure that could drive neutrophil trogocytosis. However, treatment with a CD47 blocking antibody only triggered ~10–20% trogocytosis of FaDu cells (Figure 3A), whereas treatment with Gal-9 induced up to 80% trogocytosis (Figure 3B). Further, Gal-9 treatment of a DLD-1 cell line knocked-out for CD47 (DLD-1.CD47−/−) triggered an increase of 15–20% in phagocytosis compared to the wt (Figure 3C). Thus, the change in CD47 is unlikely to fully account for the functional pro-phagocytic effects as induced by Gal-9 treatment. Indeed, when DLD-1 or FaDu cells were pre-treated with Gal-9 to induce these membrane changes before the trogocytosis experiment, trogocytosis only increased by ~10–15% compared to an ~80% increase upon co-treatment (Figure 2J, Figure 3B, Appendix A) as confirmed by linear regression analysis (β = 0.673, 95% CI −61.8–(−16.0), *p* < 0.004). Conversely, when neutrophils were pre-treated with Gal-9, trogocytosis of FaDu reached a level almost identical to that observed upon co-treatment (Figure 3B) as confirmed by linear regression analysis (β = −0.070, *p* < 0.746). Thus, the uptake of cancer cells by neutrophils upon Gal-9 treatment stemmed from a combination of Gal-9-mediated effects on cancer cells and neutrophils, with activation of neutrophils being the most prominent contributor. Of note, the established neutrophil activators, fMLP and LPS, did not increase trogocytosis (Figure 3D), despite clearly activating neutrophils as evidenced by the upregulation of CD11b and downregulation of CD62L (Appendix A). Thus, Gal-9 appears to trigger a unique activation pattern in neutrophils. Of note, Gal-9 binding was independent of the previously reported neutrophil receptor TIM-3, as neutrophils did not detectably express TIM-3 (Appendix A).

### 3.5. Galectin-9 Had Multi-Fold Stimulatory Activity on Neutrophils

In an attempt to delineate the characteristics of Gal-9-induced signaling in neutrophils, calcium flux was monitored as a key upstream signal initiator in neutrophils. Indeed, a rapid rise in the cytosolic calcium concentration [Ca^2+^] was detected within 2–3 s of treatment of freshly isolated neutrophils with Gal-9, which gradually decreased within 10 min (Figure 3E). This rapid increase in [Ca^2+^] upon Gal-9 treatment was dependent on CRD activity, as α-lactose prevented Gal-9-mediated calcium signaling (Figure 3E). Moreover, Gal-9-induced calcium flux was fully abrogated by co-treatment with BAPTA–AM but not EDTA (Figure 3F), indicating that Gal-9 triggered the release of intracellular Ca^2+^ stores.

In line with this early activation event, the treatment of neutrophils with Gal-9 also rapidly mobilized gelatinase granules as evidenced by membrane upregulation of CD11b and CD18 (Figure 3G, Appendix A). This upregulation of CD11b and co-expressed receptor CD18, together forming the MAC-1 (CR3) adhesion complex, was rapid and dose dependent. In addition to the upregulation of the MAC-1 adhesion complex, the CD11c component of the P150,95 (CR4) adhesion complex was also upregulated with similar characteristics (Figure 3G, Appendix A). In contrast, the expression of CD11a (part of complex LFA-1) was downregulated (Figure 3G, Appendix A). Treatment with Gal-9 also mobilized specific granules as evidenced by the membrane upregulation of CD15 and CD66b [36] (Figure 3G, Appendix A) and upregulated membrane expression of azurophilic granule markers CD63 (Figure 3G, Appendix A). Taken together, treatment with Gal-9 prominently activated neutrophils with rapid mobilization of intracellular calcium stores, mobilization of different granules, and upregulation of adhesion complexes.

### 3.6. Galectin-9-Mediated Activation of Neutrophils Abrogates Cancer Cell Adhesion

In line with the upregulation of the integrin complexes, MAC1/CR3 and P150,95/CR4, on neutrophils, expression of the counter receptor, ICAM-1, was upregulated on FaDu cells in mixed cultures (Figure 4A, Appendix A). Consequently, mixed cultures of neutrophils and non-adhered FaDu cells treated with Gal-9 formed large cellular aggregates within ~1 h (Figure 4B). Similarly, monocultures of isolated neutrophils treated with Gal-9 formed clusters within ~1 h of treatment (Figure 4B). In contrast, FaDu cells treated with Gal-9 did not form clusters but rapidly adhered and stretched (Figure 4B). To further characterize these clusters, cancer cell/ neutrophil co-cultures were treated with Gal-9 and cell aggregates were evaluated by flow cytometry using EpCAM and CD16 as cancer and neutrophil markers, respectively (Figure 4C). None to minimal aggregates were formed in untreated mixed cultures, but a clear formation of aggregates was detected upon treatment with Gal-9 (Figure 4C). Approximately 20% of these aggregates were composed of FaDu cells and neutrophils (CD16+/EpCAM+) (Figure 4C,D) with of up to 50% of FaDu cells being found in such aggregates in mixed cultures upon Gal-9 treatment (Figure 4C). Of note, neutrophil/FaDu aggregates were not formed when mixed cultures were treated at 4 °C and were, therefore, not due to the mere Gal-9-mediated formation of glycoconjugates (Figure 4E). Thus, treatment of neutrophils with Gal-9 triggered neutrophil activation, leading to the rapid formation of cancer cell/neutrophil clusters. These clusters remained prominently visible at longer treatment times of 16 h in Gal-9 treated but not in control treated mixed cultures (Figure 4F, G) and prevented adhesion of cancer cells to the culture plate. Upon quantification of adherent cells using MTS, a clear dose-dependent decrease in cell adhesion was detected for the Gal-9-sensitive FaDu (Figure 5A), Gal-9-resistant WiDr (Figure 5B), and a panel of different carcinoma cell lines with the exception of Caco2 (Figure 5C). Importantly, depletion of neutrophils from the leukocyte population fully abrogated the impact of Gal-9 on cancer cell adhesion (Figure 5D,E). In contrast, depletion of other cell populations, including NK-cells or monocytes, as well as treatment with isolated peripheral blood lymphocytes did not reduce cancer cell adhesion upon treatment with Gal-9 (Figure 5F). Thus, the anti-adhesive effect on cancer cells detected in these mixed culture experiments was fully attributable to neutrophils.

### 3.7. Galectin-9-Treated Neutrophils Trigger Cytotoxic Elimination of Cancer Cells

As indicated above, Gal-9 treatment of neutrophils mobilized gelatinase and specific granules, both containing potentially tumoricidal components. Correspondingly, Gal-9 treatment of pre-adhered and CFSE-labeled FaDu cells with leukocytes strongly reduced the number of remaining viable FaDu cells within 18 h (Figure 6A). In contrast, treatment with either leukocytes or Gal-9 alone did not significantly affect the monolayer of FaDu cells (Figure 6A, Appendix A). Notably, time-lapse analysis revealed a strong time-dependent reduction in FaDu cells upon Gal-9 treatment in mixed leukocyte cultures with minimal loss of viability with neutrophils alone and a maximal loss of FaDu cells of up to ~70–80% at 150 nM Gal-9 (Figure 6B, grey diamond squares). In contrast, treatment with 150 nM Gal-9 alone did not reduce the FaDu cell count in this short time frame (Figure 6B,C, white diamond squares). Similar cytotoxic activity of Gal-9-activated neutrophils was detected in a panel of carcinoma lines, comprising cell lines both sensitive and resistant to direct Gal-9 cytotoxicity with the notable exception of the Caco2 cell line that proved fully resistant to Gal-9-activated neutrophils (Figure 6D,E). Thus, the neutrophil-mediated cytotoxic activity of Gal-9 in these co-cultures did not appear to require the initial induction of cell death in cancer cells by Gal-9. Of note, at a higher E:T ratio of 10:1, neutrophil-mediated elimination of cancer cells increased by up to ~90% (Figure 6E). Taken together, Gal-9 induced the elimination of both Gal-9-resistant and Gal-9-sensitive cancer cells via the activation of neutrophils. 

## 4. Discussion

In the current study, we identified that treatment of cancer cells with Gal-9 skewed the phagocytic balance towards a pro-phagocytic phenotype by both upregulating PS and downregulating the CD47 surface expression. Moreover, Gal-9 treatment triggered prominent activation of neutrophils that, in mixed cultures with cancer cells, combined to strongly increase trogocytosis, inhibit cancer cell adhesion, and induce neutrophil-mediated cancer cell death. Notably, this innate immune-mediated anticancer activity was detected both in cell lines that were sensitive and resistant to direct Gal-9 cytotoxicity, highlighting that Gal-9-mediated neutrophil activation may provide an additional layer of anticancer activity by Gal-9. 

In this study, Gal-9 downregulated CD47 and simultaneously upregulated PS, leading to a shift in the phagocytic balance towards phagocytic elimination. The Gal-9 induced PS exposure is in line with previous studies [9]. PS is a well-established “eat me” signal, and its exposure on the cell membrane can contribute to engulfment by phagocytes [37,38]. Moreover, the downregulation of CD47 by Gal-9 may disrupt the CD47–SIRPα anti-phagocytic axis. Inhibition of CD47–SIRPα signaling in itself is a prominent therapeutic avenue that promotes phagocytosis of cancer cells by innate immune cells. For instance, CD47-blocking antibodies in combination with tumor-opsonizing antibodies significantly increased the antitumor activity of neutrophils in vitro and in vivo [39,40,41,42], whereas knockdown of CD47 increased the killing of breast cancer cells by neutrophils [43]. 

Interestingly, when neutrophils were pre-treated with Gal-9, these neutrophils still actively trogocytosed cancer cells to a much higher extent than upon pre-treatment of cancer cells, with Gal-9 before mixing with neutrophils. From these results, it is evident that Gal-9-mediated trogocytosis of cancer cells does not solely rely on the downregulation of CD47, but also stems from direct activation of neutrophils by Gal-9. Such Gal-9 mediated activation of granulocytes is in line with our previous work in which the addition of Gal-9 triggered CD11b upregulation as well as neutrophil migration [15]. Further, Gal-9 was reported to induce degranulation of neutrophils and increase their potential to phagocytose bacteria [26].

In the current study, Gal-9 induced the upregulation of various activation markers, including CD18, CD11c, CD15, and CD63, together indicating that all types of granules were mobilized. Of note, CD11b and CD18 form the adhesion complex MAC-1. MAC-1 binds to its counter receptor ICAM-1, which in mixed cultures was simultaneously induced on the surface of cancer cells. Whether this is a direct effect of Gal-9 on the surface of cancer cells was not studied. However, our previous work on bronchial epithelial cells showed that the induction of ICAM-1 was mediated by cytokines that were secreted upon Gal-9 treatment [15]. Importantly, the MAC-1 complex was reported to be essential to form conjugates between cancer cells and neutrophils and to induce the killing of antibody-opsonized cancer cells via trogocytosis [44]. Indeed, we observed a strong induction of trogocytosis of cancers cells by neutrophils upon Gal-9 treatment, which was further potentiated by treatment with the therapeutic antibody, cetuximab. 

Loss of expression of endogenous Gal-9 in cancer cells is closely associated with tumor progression and metastatic formation in several cancers [10,45]. Mechanistically, this has been attributed to the interaction of Gal-9 with CD44, whereby metastatic spread was hampered, and the number of lung metastases was reduced in a melanoma mouse model [10]. Further, expression of Gal-9 in breast cancer cells in vitro inhibited their adhesion to endothelium and/or extracellular matrix components such as collagen type I [10,11]. Based on these studies, loss of Gal-9 during cancer progression was hypothesized to modulate cell adhesion and, thereby, facilitate extravasation and metastatic spread. However, our study provides additional evidence for another hypothesis. During metastatic spread into the blood stream cancer cells first encounter neutrophils, with cancer cells expressing Gal-9, triggering neutrophil anticancer activity that inhibits metastatic progression.

Although neutrophils are perhaps best known for their pro-tumorigenic and possibly pro-metastatic activity, they also have documented anti-tumoral activity [27,28]. For instance, activated neutrophils are the driving force behind the efficacy of Bacillus Calmette–Guérin (BCG) therapy in bladder cancer [46]. Although we did not study how neutrophils eliminated cancer cells, treatment with Gal-9 induced the mobilization of all granules, which suggests that inflammatory mediators from these granules contribute to the killing of cancer cells. In addition, trogocytosis can result in cancer cell death called trogoptosis, where multiple bites lead to the disruption of membrane integrity leading to a lytic type of cell death [44]. The exact mechanism of cell death by neutrophils is the subject of ongoing studies. 

## 5. Conclusions

In conclusion, Gal-9 is a potential checkpoint regulator that shifts the phagocytic balance off cancer cells towards an “eat me” signal. This shift correlated with a strong induction of trogocytosis as well as cytotoxic activity against carcinomas by Gal-9-activated neutrophils. Therefore, Gal9 is a potential modulator of neutrophil-mediated anticancer immunity.

## Figures and Tables

**Figure 1 biomedicines-10-00066-f001:**
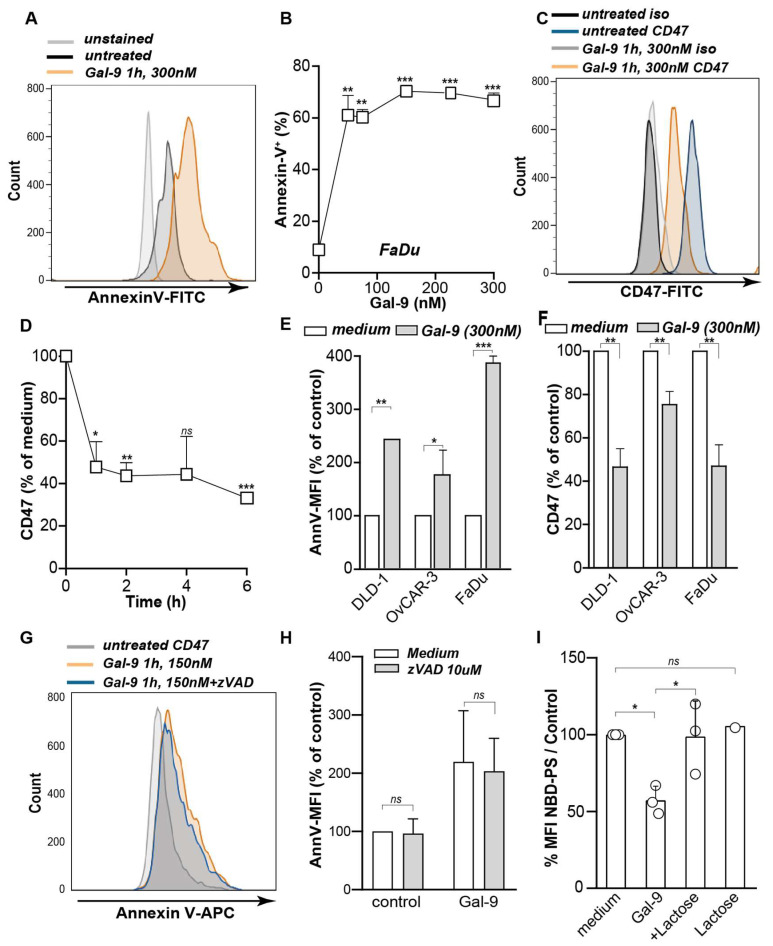
Galectin-9 induced upregulation of PS and downregulation of CD47 on carcinoma cell lines. PS expression as determined by flow cytometry on FaDu after 1 h of treatment (**A**) and (**B**) at different concentrations of Gal-9 compared to the medium control. (**C**) CD47 expression on FaDu after treatment with Gal-9 at 300 nM for 1 h and (**D**) in a time course compared to t = 0. (**E**) Gal-9-induced expression of PS and (**F**) CD47 on different carcinoma cell lines. (**G**,**H**) PS expression on Gal-9-treated FaDu in the presence of the caspase inhibitor, zVAD-fmk, as determined by flow cytometry. (**I**) NBD-PS uptake in FaDu with or without Gal-9 treatment. *ns*: not significant; * *p* < 0.05; ** *p* < 0.01; *** *p* < 0.001.

**Figure 2 biomedicines-10-00066-f002:**
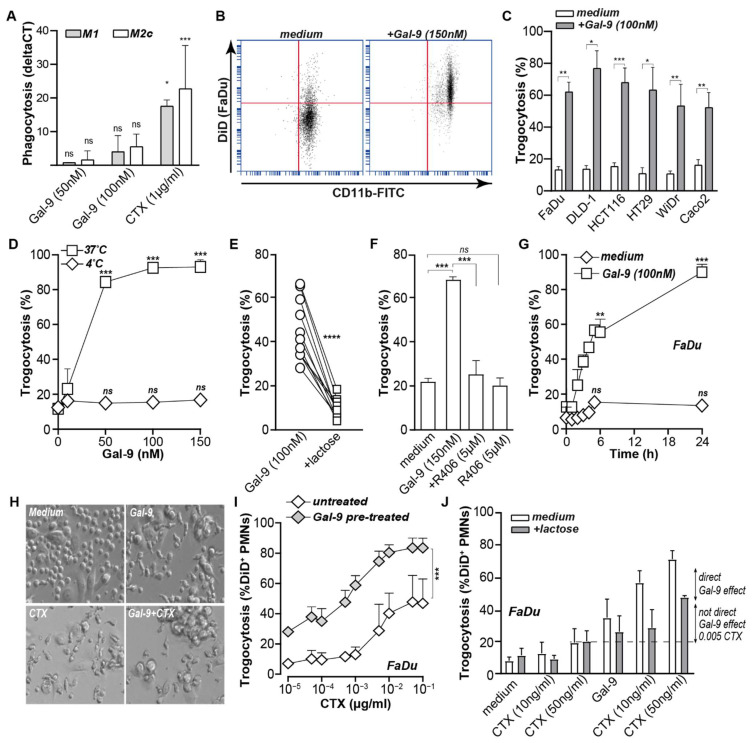
Galectin-9 triggered neutrophil but not macrophage-mediated cancer cell uptake. (**A**) Phagocytosis of untreated and Gal-9/CTX-treated FaDu with M1 and M2c macrophages as measured with flow cytometry as compared to the medium control. (**B**) Trogocytosis of untreated and Gal-9-treated FaDu in a mixed culture with neutrophils gated on the neutrophils. (**C**) Trogocytosis of Gal-9-sensitive and Gal-9-resistant carcinoma cell lines by neutrophils with Gal-9 (150 nM). (**D**) Trogocytosis of Gal-9-treated FaDu by neutrophils at 37 and 4 °C and at different Gal-9 concentrations compared to the medium control of each group. (**E**) Trogocytosis of FaDu by neutrophils with Gal-9 and Gal-9 + α-Lactose. (**F**) Inhibition of Gal-9-mediated trogocytosis of FaDu with R406. (**G**) Trogocytosis measured at different time points compared to the medium control at each time point. (**H**) Bright field microscopy images showing the morphological changes of CTX/Gal-9-treated FaDu/leukocytes/mixed culture. (**I**) Neutrophil-mediated trogocytosis of FaDu cells pre-treated with Gal-9 at different concentrations of the EGFR binding antibody CTX. (**J**) As in (**I**) in the presence of α-lactose to inhibit any residual Gal-9 bound to the cell membrane. * *p* < 0.05; ** *p* < 0.01; *** *p* < 0.001; **** *p* < 0.0001.

**Figure 3 biomedicines-10-00066-f003:**
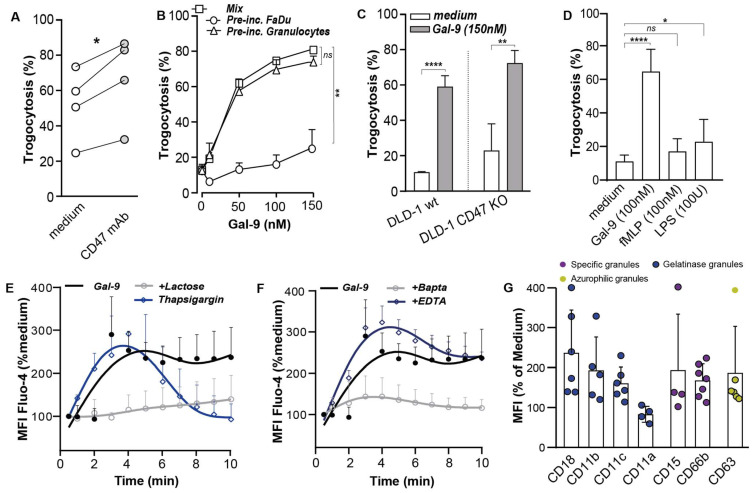
Gal-9-mediated trogocytosis can largely be attributed to neutrophil activation. (**A**) Trogocytosis of FaDu by neutrophils with CD47 mAb. (**B**) Trogocytosis of FaDu by neutrophils after FaDu/neutrophils pre-incubation with Gal-9 compared to medium control. (**C**) Gal-9 induced trogocytosis of DLD-1 wt and DLD-1 CD47 KO by neutrophils. (**D**) Trogocytosis of FaDu by neutrophils with Gal-9/fMLP/LPS. (**E**,**F**) (Ca^2+^) Flux in Gal-9-treated neutrophils within a time range of 0–10 min measured using flow cytometry in the presence or absence of calcium inhibitors. (**G**) Expression of different granule markers on Gal-9-treated neutrophils. Azurophilic granules (CD15), specific granules (CD63 and CD66b), and gelatinase granules (CD11b and CD18). * *p* < 0.05; ** *p* < 0.01; **** *p* < 0.0001.

**Figure 4 biomedicines-10-00066-f004:**
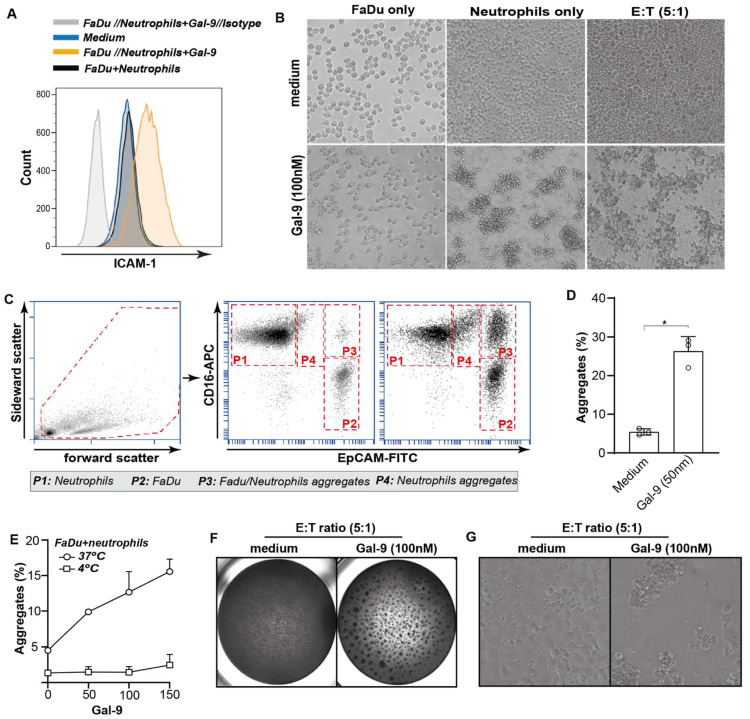
Gal-9 mediated the aggregation of FaDu and neutrophils. (**A**) ICAM-1 expression on FaDu as determined by flow cytometry in a mixed culture with neutrophils and Gal-9. (**B**) Brightfield microscopy images of FaDu, neutrophils, and mixed cultures with and without Gal-9 treatment (100 nM). (**C**,**D**) Aggregation of FaDu and neutrophils with Gal-9 treatment as measured by flow cytometry. * *p* < 0.05. (**E**) Quantification of FaDu–neutrophil aggregates at 0 and 37 °C at different Gal-9 concentrations. (**F**,**G**) Brightfield microscopy pictures of untreated and Gal-9-treated mixed cultures of FaDu and neutrophils.

**Figure 5 biomedicines-10-00066-f005:**
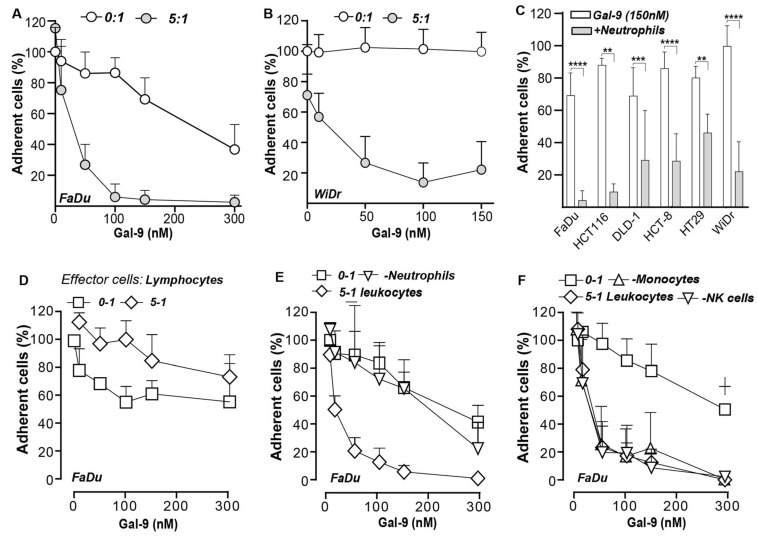
Gal-9 inhibits the adhesion of carcinoma cell lines in mixed cultures with leukocytes. (**A**) Adhesion of (**A**) FaDu and (**B**) WiDr in a mixed culture with neutrophils at different Gal-9 concentrations. (**C**) Adhesion of different carcinoma cell lines in a mixed culture with neutrophils treated with Gal-9 (150 nM); ** *p* < 0.01; *** *p* < 0.001; **** *p* < 0.0001. Adhesion of FaDu at different Gal-9 concentrations in a mixed culture with (**D**) lymphocytes, (**E**) leukocytes depleted for neutrophils, and (**F**) leukocytes depleted for NK cells and monocytes.

**Figure 6 biomedicines-10-00066-f006:**
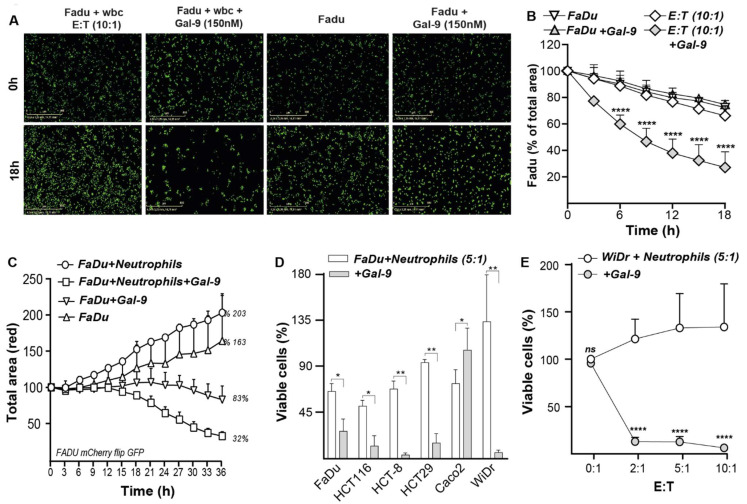
Gal-9 mediated the cytotoxic elimination of different carcinoma cell lines by leukocytes. (**A**) Fluorescence microscopy images of untreated and Gal-9-treated CFSE-labeled FaDu with and without leukocytes after 18 h. (**B**) Viability of untreated and Gal-9 (150 nM)-treated FaDu in a mixed culture with leukocytes after 18 h and (**C**) after 36 h of incubation compared to the medium control. (**D**) Cytotoxic removal of different carcinoma cell lines by neutrophils with Gal-9. (**E**) Viability of the Gal-9-resistant cell line WiDr in a mixed culture with neutrophils and Gal-9. Gal-9-treated groups compared to untreated groups at the same E:T ratio. * *p* < 0.05; ** *p* < 0.01; **** *p* < 0.0001.

## Data Availability

Not applicable.

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
