# Peer review of "Galectin-9 Triggers Neutrophil-Mediated Anticancer Immunity"

_biomedicines, 2021, doi:10.3390/biomedicines10010066_

Round 1

Reviewer 1 Report

This manuscript provided evidence that Gal-9 sensitized carcinoma cells to immune recognition by increasing PS-exposure within minutes, and decreasing surface expression of  CD47. The authors proposed that treatment with Gal-9 may activated neutrophils, leading to cancer cell trogocytosis, and
neutrophil-mediated cytotoxic activity in cancer cells. suggesting Gal-9 in neutrophil-mediated anticancer immune responses. 

Although the authors proposed a possible new function of Gal-9, several serious flaws was throughout this version of manuscript, especially authors did not show the number of donor of each experiment and did not reveal statistic results of most of data, which makes reviewers unable to judge the results. Thus, I hope the authors can do the corrections before next submission.

Other suggestion:

  1. The ref 25 is missed in the article
  2. The meaning of Page 2 line 70-72 is not clear.
  3. Why authors used FaDu cells for most experiments.
  4. The scale of histogram should be shown.
  5. The representative of gating on positive population should be provided (such as Fig1B, 1D).
  6. What not use 2D scatter plot for AnnexinV and CD47.     
  7. Fig 1I has two Lectose treatment.
  8. Dose Gal-9 induce neutrophil activation in vivo? it seems worth to test on mouse model. 

Author Response

Hereby we submit our revised manuscript “Galectin-9 triggers neutrophil-mediated anticancer immunity”

by Natasha Ustyanovska Avtenyuk, Ghizlane Choukrani, Emanuele Ammatuna, Toshiro Niki, Ewa Cendrowicz Król, Harm Jan Lourens, Gerwin Huls, Valerie R Wiersma and Edwin Bremer

for publication in Biomedicines.

We have revised our manuscript according to the reviewer’s suggestions and hope that our manuscript now meets the criteria for publication in Biomedicines. A point-by-point response to the reviewers’ comments is detailed below.

Point-by-point reply to reviewers’ comments:

Reviewer #1

Comments and Suggestions for Authors

This manuscript provided evidence that Gal-9 sensitized carcinoma cells to immune recognition by increasing PS-exposure within minutes, and decreasing surface expression of CD47. The authors proposed that treatment with Gal-9 may activated neutrophils, leading to cancer cell trogocytosis, and

neutrophil-mediated cytotoxic activity in cancer cells. suggesting Gal-9 in neutrophil-mediated anticancer immune responses.

Although the authors proposed a possible new function of Gal-9, several serious flaws was throughout this version of manuscript, especially authors did not show the number of donor of each experiment and did not reveal statistic results of most of data, which makes reviewers unable to judge the results. Thus, I hope the authors can do the corrections before next submission.

Response:

In the revised manuscript we have included statistics throughout the figures and detailed in the manuscript. The number of donors for each panel was at least 3 donors/condition.

Other suggestion:

Comment 1

The ref 25 is missed in the article

Response to Comment 1

The reference has been added to the revised manuscript.

Comment 2

The meaning of Page 2 line 70-72 is not clear.

Response to Comment 1

We included this sentence to clarify that instead of using the term PMN (encompassing neutrophils but also basophils and eosinophils), we would be using the term neutrophils throughout the remainder of the manuscript.

Comment 3

Why authors used FaDu cells for most experiments.

Response to Comment 3

The cell line FaDu was chosen as an exemplary cell line for the assessment of Gal-9-mediated activity. However, at each key experimental point, the findings obtained for FaDu were confirmed in a panel of cell lines to ensure the solidity of our experimental findings.

Comment 4

The scale of histogram should be shown.

Response to Comment 4

We have added the histogram scaling in the figures.

Comment 5

The representative of gating on positive population should be provided (such as Fig1B, 1D).

Response to Comment 5

Since the entire cell population is positive for PS and CD47, the only gating performed was done on the live cells in the forward/sideward scatter plot.

Comment 6

What not use 2D scatter plot for AnnexinV and CD47.    

Response to Comment 6

We did not use 2D scatter plots since the two fluorescent agents used were both in the FL1 fluorescent channel. Notably, the increase in PS and the decrease in CD47 is uniform, with a shift in the intensity of staining of the entire population. Therefore, although of interest, we do not believe a 2D scatter plot would yield additional information.

Comment 7

Fig 1I has two Lectose treatment.

Response to Comment 7

We apologize for the ambiguity in the description. We have clarified the meaning, with one condition being treated with alpha-lactose alone and the second condition being treated with Galectin-9 and alpha-lactose.

Comment 8

Dose Gal-9 induce neutrophil activation in vivo? it seems worth to test on mouse model.

Response to Comment 8

We agree that evaluating in vivo effects of Gal-9 on neutrophils is of great interest (e.g., by performing intra-vital microscopy for neutrophil-mediated phagocytosis). However, we believe that such studies are beyond the scope of the current manuscript, also in view of the timeline of the special issue.

Reviewer 2 Report

The manuscript titled "Galectin-9 triggers neutrophil-mediated anticancer immunity" contain interesting data about Galectin-9-mediated activation of neutrophils in view of trogocytosis and augmented antibody-dependent cellular phagocytosis of cancer cells. In general this data are original and interesting. Obtained results and its interpretation making no doubt. This manuscript appropriate for publication in Biomedicines.

Author Response

Hereby we submit our revised manuscript “Galectin-9 triggers neutrophil-mediated anticancer immunity”

by Natasha Ustyanovska Avtenyuk, Ghizlane Choukrani, Emanuele Ammatuna, Toshiro Niki, Ewa Cendrowicz Król, Harm Jan Lourens, Gerwin Huls, Valerie R Wiersma and Edwin Bremer

for publication in Biomedicines.

We have revised our manuscript according to the reviewer’s suggestions and hope that our manuscript now meets the criteria for publication in Biomedicines. A point-by-point response to the reviewers’ comments is detailed below.

Reviewer 2

Comments and Suggestions for Authors

The manuscript titled "Galectin-9 triggers neutrophil-mediated anticancer immunity" contain interesting data about Galectin-9-mediated activation of neutrophils in view of trogocytosis and augmented antibody-dependent cellular phagocytosis of cancer cells. In general this data are original and interesting. Obtained results and its interpretation making no doubt. This manuscript appropriate for publication in Biomedicines.

Response

We thank the reviewer for his assessment that the manuscript is suitable for publication in Biomedicines.

Round 2

Reviewer 1 Report

The revised manuscript have significant improvement, and I thank the finding will raise interests for readers in related field. However, before publication, several description of data still need to carefully revised throughout the article.   

For example:

  1. The meaning of four star "****" did not provide.
  2. The comparison need to proper clarify in the legend, whether the differences are between dosages or groups.  
  3. In addition, several comparisons are more meaningful to know whether there is difference between groups, such as Fig. 2I and Fig. 3B.
  4. Figure 4C, the labels of sample are missing.
  5. Figure 6A. Is Fadu + WBC correct? and in Figure 6B, the labels "+Gal(150nM)" and + "Gal-9" are difficult to understand the differences.   

Author Response

We thank the reviewer for careful reading of the manuscript constructive remarks. We have taken the comments onboard to improve the description of data and clarify the manuscript. Please find bellow a detailed point-by-point response to all comments.

Comment 1

The meaning of four star "****" did not provide.

Response to Comment 1

The meaning of four stars has been added to the revised manuscript.

Comment 2

The comparison need to proper clarify in the legend, whether the differences are between dosages or groups. 

Response to Comment 2

The comparisons have been clarified in the figures with a line to indicate which groups have been compared in the statistical analysis.

Comment 3

In addition, several comparisons are more meaningful to know whether there is difference between groups, such as Fig. 2I and Fig. 3B.

Response to Comment 3

The comparison between the groups in Fig. 2I and Fig. 3B have been added to the figures.

Comment 4

Figure 4C, the labels of sample are missing.

Response to Comment 4

The label of Figure 4C is included in the figure legend (…(C-D) Aggregation of FaDu and neutrophils…), Lines 395-396.

Comment 5

Figure 6A. Is Fadu + WBC correct? and in Figure 6B, the labels "+Gal(150nM)" and + "Gal-9" are difficult to understand the differences.  

Response to Comment 5

Indeed, Fadu + wbc is correct and serves to indicate that a mixed culture was performed. The description of experimental conditions in figure 6B has been adapted for clarity in the revised manuscript.
